# Impact of Illness Severity and Interventions on Successful Weaning from Nasal CPAP in Very Preterm Neonates: An Observational Study

**DOI:** 10.3390/children9050673

**Published:** 2022-05-06

**Authors:** I-Ling Chen, Hsiu-Lin Chen

**Affiliations:** 1Department of Respiratory Therapy, College of Medicine, Kaohsiung Medical University, No. 100, Shih-Chuan 1st Road, San Ming District, Kaohsiung 80708, Taiwan; ilchen@kmu.edu.tw; 2Department of Pediatrics, Kaohsiung Medical University Hospital, No. 100, Tzyou 1st Road, San Ming District, Kaohsiung 807, Taiwan

**Keywords:** nasal CPAP, very preterm neonates, weaning, postmenstrual age, body weight, nutrition

## Abstract

This study aims to identify clinical variables that could affect successful weaning from nasal continuous positive airway pressure (NCPAP) in very preterm infants. Infants born at a gestational age (GA) of <32 weeks were retrospectively enrolled. Weaning from NCPAP was initiated when the infants were clinically stable. In the univariate analysis, GA, birth weight, body weight (BW) z-score at the time of successful NCPAP weaning, intubation, total duration of intubation, respiratory distress syndrome grade, APGAR score at the 1 and 5 min, initial shock, anemia, bronchopulmonary dysplasia, number of blood transfusions, total duration of dopamine use, administration of more than two doses of surfactant, use of aminophylline, use of a diuretic, and total duration of total parenteral nutrition were significantly associated with postmenstrual age (PMA) at the time of successful NCPAP weaning. Multivariate analysis showed that the total duration of intubation, bronchopulmonary dysplasia, and administration of more than two doses of surfactant were positively associated with PMA at the time of successful NCPAP weaning. A reverse association was noted between BW z-score and PMA at the time of successful NCPAP weaning. Sufficient nutrition and avoidance of further ventilator-induced lung injury could decrease NCPAP duration in very preterm infants.

## 1. Introduction

Due to the notable incidence of respiratory complications attributed to mechanical ventilation, the use of alternative respiratory support has become common in the clinical management of preterm infants. Nasal continuous positive airway pressure (NCPAP) is a non-invasive ventilation technique that improves gas exchange by delivering constant positive pressure to recruit semi-collapsed alveoli and preventing atelectasis during expiration [1]. NCPAP could support lung growth [1,2], decrease work of breathing [1,2], and reduce the need for intubation [3,4,5] and exogenous surfactant administration [5]. Furthermore, using NCPAP as the primary respiratory support in very-low-birth-weight preterm infants with respiratory distress syndrome (RDS) lowered the morbidity of bronchopulmonary dysplasia (BPD) [6]. Because of associated complications, including nasal trauma, gastric distension, pneumothorax, intraventricular hemorrhage, and patient discomfort [1], minimizing NCPAP duration may be beneficial. However, its premature cessation could lead to apnea and increased demand for respiratory support. Thus, it is crucial to predict and determine the proper timing of NCPAP weaning.

Although the use and benefits of CPAP have been investigated, an agreement on the starting point for NCPAP weaning and information about factors that could determine whether the infants could be weaned from NCPAP remains elusive. Thus, this study aimed to identify the medical conditions and interventions that could affect successful weaning from NCPAP in preterm infants born at a gestational age (GA) of <32 weeks.

## 2. Methods

### 2.1. Study Participants

This retrospective study included infants born at a GA of <32 weeks requiring positive pressure ventilation support and were admitted to the neonatal intensive care unit (ICU) of Kaohsiung Medical University Hospital in Taiwan between June 2013 and March 2016. The experimental protocol was approved by the Institutional Review Board (IRB) of Kaohsiung Medical University Hospital (IRB number, KMUHIRB-SV(I)-20200028; date, 12 June 2020). Informed consent was not obtained because our research was a retrospective review using medical records data without contacting subjects’ parents. The exclusion criteria were any major birth defects/chromosomal abnormalities, failed weaning at the first attempt, and death during hospitalization. The condition of those who failed the first weaning attempt was worse than that of those who successfully weaned from NCPAP. Therefore, it was not suitable to compare these groups. We retrospectively recorded the primary and antenatal data (GA, birth weight (BBW), sex, type of birth, APGAR scores, the Neonatal Therapeutic Intervention Scoring System (NTISS, a scoring system used to indicate disease severity for neonates who need intensive care) [7], antenatal use of steroids, preeclampsia, and chorioamnionitis), duration of intubation and NCPAP use, postmenstrual age (PMA), current weight (BW), and BW z-score [8] at the time of NCPAP weaning, associated medical condition (RDS grade, initial shock, clinical sepsis, pneumothorax, anemia, BPD, necrotizing enterocolitis, intraventricular hemorrhage, and retinopathy of prematurity requiring laser therapy), details of medication use (use of surfactant, dopamine, aminophylline, and diuretics), and supplementation with total parenteral nutrition.

The initial shock was defined as blood pressure < GA within 24 h after birth. Infants were diagnosed as anemic with the need for packed red blood cells if the hematocrit level was <22% at room air, <30% in infants requiring CPAP, or <35% in intubated and mechanically ventilated infants [9]. BPD was defined as the need for supplemental oxygen or positive-pressure ventilatory support (including invasive positive ventilation and NCPAP) at 28 days. Intratracheal dose of modified bovine surfactant extract (Survanta, 100 mg/kg, Ross/Abbott Laboratories, Columbus, OH, USA) was used in Kaohsiung Medical University Hospital.

The nutritional strategy for preterm infants in our hospital includes early trophic feeding by an orogastric tube to establish early enteral nutrition and early initiation of parenteral nutrition with glucose, amino acid (starting with 2–3 g/kg/day), and fat. If the infant tolerates enteral nutrition, the feed volume is advanced by increments of 10–20 mL/kg/day. The target caloric intake for preterm infants is around 110–130 kcal/kg/day to gain good weight during hospitalization.

### 2.2. Methods of NCPAP Weaning

Weaning from NCPAP was initiated when the very preterm infant was clinically stable (which was defined as infants on room-air NCPAP with no apnea for at least 48 h). The major NCPAP weaning method at our hospital is cyclic NCPAP with room air, with a gradual increase in time-off NCPAP in 3-h intervals. For example, when the infant was clinically stable, we attempted to remove NCPAP for an hour in the 3-h interval with the 2 h using NCPAP. If the infant could tolerate without NCPAP (defined as no tachypnea, no oxygen desaturation, and no increase in apnea frequency), we increased the time in room air next day, (e.g., 2 h without NCPAP and 1 h using NCPAP in the 3-h interval). Successful NCPAP weaning was defined as stability in room air without any respiratory support for 7 days.

### 2.3. Statistical Analysis

Descriptive data of the subjects’ clinical variables are presented as mean ± standard deviation (SD) or as *n* (%). Linear regression, univariate, and multivariate analyses were conducted to identify the association between clinical variables and PMA at the time infant was successfully weaned off NCPAP. All analyses were performed using the Statistical Package for Social Science (version 24 (Windows); SPSS Inc., Chicago, IL, USA). Differences or associations were considered statistically significant at *p* < 0.05.

## 3. Results

### 3.1. Clinical Characteristics of the Participants

Sixty-one very preterm infants were initially enrolled. Twelve were excluded: six died during hospitalization, five used high-flow nasal cannula (HFNC), and one was an outlier who failed weaning at the first attempt and ultimately ended up with a tracheostomy. We excluded the infants who used HFNC as the NCPAP weaning method due to the small sample size. Therefore, 49 patients were eligible for the study, including 24 boys and 25 girls, with a GA of 28.1 ± 2.1 weeks and BBW of 1072.7 ± 267.0 g. Nineteen infants were intubated, and the mean total duration of intubation was 21.8 ± 26.0 days. The duration of respiratory support was 41.6 ± 31.4 days, with a PMA at 34.0 ± 3.0 weeks and BW of 1809.5 ± 628.2 g at the time of successful NCPAP weaning. The primary and antenatal variables of the study population are shown in Table 1 and Table 2, respectively.

### 3.2. Association between Clinical Characteristics and Successful NCPAP Weaning

We performed correlation and regression analyses to determine potential factors that may affect successful NCPAP weaning. As shown in Figure 1, GA, BBW, and BW z-score were negatively associated with PMA at successful NCPAP weaning. The beta values of the regression line were −0.74 and −0.07 for GA and BBW, respectively. Therefore, the duration of NCPAP decreased by 0.74 weeks for every week increase in GA and by 0.07 g for every gram increase in BBW. Moreover, a negative association was noted between PMA and the BW z-score at successful NCPAP weaning. The beta value of the regression line was −0.85.

#### NCPAP, Nasal CPAP; PMA, Postmenstrual Age

Table 3 shows the univariate analysis of the association of clinical variables with PMA at the time of successful NCPAP weaning. In addition to GA, BBW, and BW z-score at successful NCPAP weaning, APGAR score at the 1 and 5 min was negatively associated with PMA at the time of successful NCAP weaning. In contrast, intubation, duration of intubation, RDS grade, initial shock, anemia, BPD, number of blood transfusions, total duration of dopamine use, administration of more than two doses of surfactant, use of aminophylline, total duration of aminophylline use, use of a diuretic, total duration of diuretic use, and total duration of total parenteral nutrition were positively associated with PMA at the time of successful NCPAP weaning.

Among the clinical variables with a significant association in the univariate analysis, the total duration of intubation, BPD, and administration of more than two doses of surfactant were positively associated with PMA at the time of successful NCPAP weaning, whereas a reverse association was observed between BW z-score and PMA at the time of successful NCPAP weaning in the multivariate analysis (Table 4). No other significant associations were found in the multivariate analysis.

## 4. Discussion

NCPAP is a widely used and effective non-invasive method for managing respiratory distress in infants because its use is associated with lower pulmonary morbidity [10]. This method is recommended as the primary ventilatory support in very-low-birth-weight preterm infants [11]. Despite these benefits, complications due to NCPAP use are non-negligible. Furthermore, early or delayed NCPAP weaning could cause harm to the infant. It is critical to determine the appropriate timing to wean infants from NCPAP. Therefore, we aimed to identify the clinical variables that may influence PMA at successful NCPAP weaning. In this study, we demonstrated that very preterm infants born at GA < 32 weeks can be successfully weaned from NCPAP at a PMA of 34.0 ± 3.0 weeks and a BW of 1809.5 ± 628.2 g. Consistent with a previous study [12], we found an inverse association between GA and BBW and PMA at the time of successful NCPAP weaning. A similar scenario was observed in the APGAR score at the 1st and 5th minute. These findings suggest that prematurity and poor health conditions are related to an increased duration of NCPAP use.

A negative correlation was observed between BW z-score and PMA at the time of successful NCPAP weaning, which suggested that a better BW z-score is associated with a shorter weaning period of NCPAP. Adequate nutritional management is particularly crucial in the neonatal period of extremely low-birth-weight infants [13]. Nutritional interventions early in life may decrease the risk of BPD in very preterm infants [14]. Furthermore, those who received aggressive nutrition and early NCPAP had similar lung function and rates of lower respiratory tract infection and hospital admission as term infants [15]. These findings support the significance of nutrition in very preterm infants during neonatal ICU stay and confirm that raising very preterm infants to adequate weight gain with adequate nutrition could shorten their dependence on respiratory support. According to the European Society of Pediatric Gastroenterology, Hepatology, and Nutrition, recommended enteral nutrient intakes for very preterm infants are 110–135 kcal/kg/day [16]. In this study, those infants who required parenteral and enteral nutrition received around 110–120 kcal/kg/day while they were clinically stable.

Shock, which is a state of decreased oxygen delivery to the tissues, is an independent predictor of early neonatal mortality, especially in very preterm infants within 24 h after birth [17]. Dopamine is commonly used to raise blood pressure in hypotensive infants. In the current study, the total duration of dopamine use was positively associated with higher PMA at the time of successful NCPAP weaning, suggesting those very preterm infants with worse conditions, such as shock or sepsis, within 24 h after birth needed a longer duration of respiratory support. In addition, most infants with shock require ventilation, and higher oxygen saturation is demanded to decrease mortality [18]. Additionally, this observation may help explain the positive association between the duration of dopamine use and PMA at NCPAP weaning.

RDS, a condition caused by surfactant deficiency, commonly occurs in very preterm infants during the first 12–48 hours of postnatal life. Exogenous surfactant administration during RDS management improves pulmonary gas exchange and reduces the requirement for mechanical ventilation and the risk of BPD [19]. Rescue surfactant has been recommended to prevent CPAP failure [20,21]. Multiple doses are sometimes required in ongoing RDS [21]. BPD and RDS severity were positively associated with PMA at weaning in our study. Moreover, the number of surfactant doses, rather than surfactant use, was associated with PMA at the time of successful NCPAP weaning. These findings suggest that lung immaturity plays a critical role in the need for respiratory support. Theoretically, prenatal steroid administration reduces RDS severity and improves the response to surfactant treatment. In contrast to a previous study [22], we did not observe any associations with PMA at the time of successful weaning from NCPAP, as noted in the study by Rastogi et al. [12]. This discrepancy may result from a lower percentage of antenatal steroid administration (49%) in this study, whereas other studies had higher rates of antenatal steroid administration (70–90%) [22,23,24]. Diuretics are commonly used to manage BPD to reduce fluid accumulation and assist in the reabsorption of excessive interstitial edema, thereby improving lung function [25]. This finding suggested that very preterm infants with poor pulmonary conditions may require a longer time to wean off NCPAP.

Decreased oxygen delivery and increased work of breathing induced by anemia could delay NCPAP weaning [12]. Consistent with this observation, our findings showed that the number of blood transfusions was positively associated with PMA at the time of successful NCPAP weaning. This scenario may further indicate that the severity of anemia influenced the need for respiratory support.

Apnea is a developmental disorder in premature infants. Its incidence is associated with GA and BBW [26]. NCPAP and methylxanthine therapy, including caffeine and aminophylline, are effective treatments for apnea [26]. Aminophylline was used exclusively in our hospital for apnea of prematurity because caffeine was not available in Taiwan at the time of this study. Infants with apnea of prematurity were administered aminophylline continuously until they became free of apneic episodes. Therefore, this practice may account for the positive association between the total duration of aminophylline and the duration of NCPAP use.

Pneumothorax [22,27] and intraventricular hemorrhage [28] are associated with CPAP failure in the treatment of RDS in preterm infants. Moreover, the occurrence of pneumothorax, necrotizing enterocolitis, and intraventricular hemorrhage is significantly associated with a greater PMA at the time of successful NCPAP weaning in very preterm infants with GA < 32 weeks [12]. Although these comorbidities and medication usage were not significant factors in our study, we could not eliminate their influence on the duration of NCPAP use because of the low incidence of these factors in our study population.

In the multivariate analysis, the total duration of intubation, BPD, and the use of more than two doses of surfactant were positively related to PMA at the time of successful NCPAP weaning. These findings indicated that the poor pulmonary conditions resulted in a longer period of NCPAP support. It is not surprising that very preterm infants with poor pulmonary conditions require prolonged respiratory support. As intubation can further contribute to ventilator-induced lung injury, prolonged intubation may delay the time of NCPAP weaning.

To date, a consensus regarding the weaning modality for NCPAP in medical centers is lacking. Weaning methods include the sudden removal of NCPAP with or without oxygen supplementation, gradual increase in time off NCPAP, gradual reduction of NCPAP pressure, or a combination of these methods. The success of weaning was either varied [29,30,31] or comparable [32,33] between different weaning methods. Over the recent years, humidified HFNC, which delivers humidified gas flow at 2–8 L/min and aims to reduce work of breathing and improve ventilation [34], has been introduced. The duration of NCPAP use could be reduced in combination with humidified HFNC without increasing the overall duration of non-invasive respiratory support and the risk of nasal lesions and BPD in very preterm infants with RDS [35]. However, the successful weaning rate was similar between the use of humidified HFNC and the gradual reduction of NCPAP pressure [36]. We excluded very preterm infants who used humidified HFNC in this study because of the very small number of these cases at our institution. Therefore, the limitation of this study was its inability to evaluate the association between different weaning methods and the clinical parameters we examined. Despite the retrospective nature of the study, current findings may help clinicians be aware of factors affecting the duration of NCPAP use in very preterm infants, thereby improving clinical management and cost-effectiveness and reducing the chance of NCPAP weaning failure. Certainly, a larger sample size and further prospective studies are needed, and multicenter trials are encouraged.

## 5. Conclusions

The current study showed that the BW z-score could shorten the time of NCPAP support in very preterm infants. The total duration of intubation, BPD, and the use of more than two doses of surfactant were positively related to PMA at the time of successful NCPAP weaning. These findings reveal the importance of nutrition and that further avoidance of ventilator-induced lung injury could decrease the duration of NCPAP use in very preterm infants. Hence, identifying and effectively managing these factors may help shorten very preterm infants’ dependence on respiratory support and predict the appropriate time for NCPAP weaning.

## Figures and Tables

**Figure 1 children-09-00673-f001:**
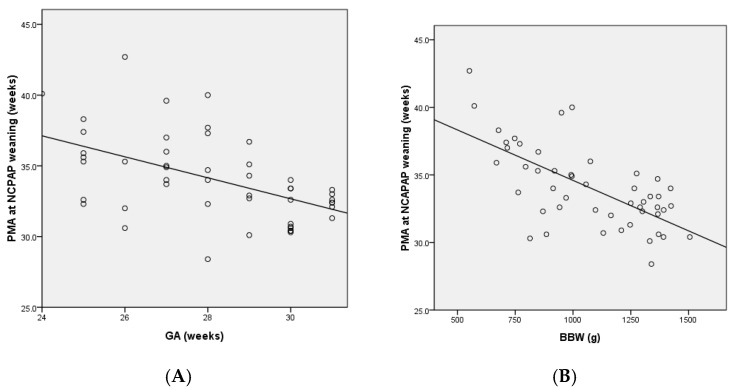
Linear regression analysis of the association between PMA at successful NCPAP weaning and clinical variables. PMA at successful NCPAP weaning was negatively associated with GA (*p* = 0.001) (**A**), BBW (*p* < 0.001) (**B**), and BW z-score (*p* = 0.01) (**C**).

**Table 1 children-09-00673-t001:** Primary and antenatal variables of the study population.

Variable	*n* = 49
Male	24 (49)
GA (weeks)	28.14 ± 2.092
BBW (g)	1072.69 ± 266.978
Cesarean section	20 (40.8)
APGAR score at 1 min	5 ± 1.879
APGAR score at 5 min	6.92 ± 1.579
NTISS	19.35 ± 6.939
Prenatal steroid use	24 (49)
Preeclampsia	11 (22.4)
Chorioamnionitis	0 (0)

Data are presented as mean ± SD or as *n* (%). GA, gestational age; BBW, birth body weight; NTISS, neonatal therapeutic intervention scoring system.

**Table 2 children-09-00673-t002:** Clinical variables of the study population during hospitalization.

Variables	*n* = 49
Respiratory support	
Intubation during hospitalization	19 (38.8)
Total duration of intubation (days)	21.8 ± 26.0
Total duration of respiratory support (days) *	41.6 ± 31.4
PMA at NCPAP weaning (weeks)	34.0 ± 3.0
BW at NCPAP weaning (g)	1809.5 ± 628.2
BW z-score at NCPAP weaning (g)	−0.8 ± 1.3
Medical condition	
RDS grade ≥ 3	21 (42.9)
Pneumothorax	1 (2)
Initial shock	12 (24)
Clinical sepsis	17 (34.7)
Anaemia	31 (63.3)
NEC (both medical and surgical)	4 (8.2)
IVH, any grade	12 (24.5)
ROP requiring laser therapy	3 (6.1)
BPD	27 (55.1)
Medical treatment during hospitalization	
Blood transfusion (times)	4.4 ± 3.6
Dopamine use	9 (18.4)
Total duration of dopamine use (days)	17.8 ± 20.0
Surfactant use	12 (24.5)
Exogenous surfactant use ≥2 times	4 (8.2)
Aminophylline use	20 (40.8)
Total duration of aminophylline use (days)	56.1 ± 24.4
Diuretic use	25 (51)
Total duration of diuretic use (days)	20.9 ± 14.8
Nutrition supplementation	
Total duration of TPN (days)	29.2 ± 22.5

Data are presented as mean ± SD or as *n* (%). BPD, bronchopulmonary dysplasia; BW, body weight; IVH, intraventricular hemorrhage; NCPAP, nasal CPAP; NEC, necrotizing enterocolitis; PMA, postmenstrual age; RDS, respiratory distress syndrome; ROP, retinopathy of prematurity; TPN, total parenteral nutrition. * The total duration of respiratory support included intubation, NIPPV, NCPAP, and O_2_ support.

**Table 3 children-09-00673-t003:** Univariate analysis of the association between clinical variables and PMA at the time of successful NCPAP weaning.

Variables	Coefficient	*p*-Value
Sex	−0.517	0.55
GA	−0.742	<0.001
BBW	−0.007	<0.001
Caesarean section	−0.223	0.8
APGAR score at 1 min	−0.857	<0.001
APGAR score at 5 min	−0.92	<0.001
NTISS	0.074	0.24
Prenatal steroid use	−0.386	0.66
Preeclampsia	0.514	0.62
Intubation	3.818	<0.001
Total duration of intubation	0.096	<0.001
RDS grade	1.014	0.03
BW z-score at NCPAP weaning	−0.845	0.01
Pneumothorax	1.585	0.61
Initial shock	2.697	0.01
Clinical sepsis	1.603	0.08
Anaemia	3.499	<0.001
NEC	2.916	0.063
IVH	1.185	0.24
ROP requiring laser therapy	1.428	0.11
BPD	4.053	<0.001
Blood transfusion	0.556	<0.001
Dopamine use	2.107	0.057
Total duration of dopamine use	0.108	0.01
Surfactant use	1.891	0.058
Administration of more than two doses of exogenous surfactant	3.352	0.031
Aminophylline use	2.945	<0.001
Total duration of aminophylline use	0.058	<0.001
Diuretic use	2.852	0.001
Total duration of diuretic use	0.107	<0.001
Total duration of TPN	0.086	<0.001

BBW, birth body weight; BPD, bronchopulmonary dysplasia; GA, gestational age; IVH, intraventricular hemorrhage; NCPAP, nasal CPAP; NEC, necrotizing enterocolitis; NTISS, neonatal therapeutic intervention scoring system; PMA, postmenstrual age; RDS, respiratory distress syndrome; ROP, retinopathy of prematurity; TPN, total parenteral nutrition.

**Table 4 children-09-00673-t004:** Multivariate analysis of the association between clinical variables and PMA at successful weaning from NCPAP.

Variables	Coefficient	SD	*p*-Value
GA	0.002	0.281	>0.99
BBW	0.001	0.002	0.43
APGAR score at 1 min	−0.322	0.291	0.28
APGAR score at 5 min	0.044	0.336	0.9
Total duration of intubation	0.084	0.032	0.01
RDS grade	−0.253	0.324	0.44
BW z-score at NCPAP weaning	−0.943	0.244	0.001
Initial shock	−0.579	0.858	0.51
BPD	2.875	0.902	0.003
Blood transfusion	−0.287	0.19	0.14
Total duration of dopamine use	−0.066	0.043	0.13
Administration of more than two doses of exogenous surfactant	3.011	1.36	0.034
Total duration of aminophylline use	0.021	0.017	0.23
Total duration of diuretic use	0.005	0.032	0.89
Total duration of TPN	0.029	0.027	0.3

BBW, birth body weight; BPD, bronchopulmonary dysplasia; GA, gestational age; NCPAP, nasal CPAP; PMA, postmenstrual age; RDS, respiratory distress syndrome; TPN, total parenteral nutrition.

## Data Availability

The data presented in this study are available on request from the corresponding author.

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
