# Peer review of "Impact of Illness Severity and Interventions on Successful Weaning from Nasal CPAP in Very Preterm Neonates: An Observational Study"

_children, 2022, doi:10.3390/children9050673_

Round 1

Reviewer 1 Report

The manuscript by I-Ling Chen et al. entitled “Impact of Medical Condition and Interventitions on Successful Weaning from Nasal CPAP in Very Preterm Neonates: an observational study” focuses on an interesting topic in perinatal medicine. Particularly, authors attempted to evaluate clinical variables that could affect successful weaning from NCPAP in very preterm infants. 

Although the aims of the study are relevant, the manuscript has some major flaws.

Abstract

  • The section needs to be revised and better clarify the study design

Method section: 

  • Material and Method section is difficult to read, especially the "study population" part. The section needs to better guide and clarify the design of the study

  • The classification of groups should be made more easily interpretable. For example could be better starting directly with the eligible patients of the study instead of infants enrolled.  

Result section:

  • The result section is difficult to read and does not provide clear and concise data on the results of the study

It would be better if you would kindly revise the manuscript by a native English speaker

Author Response

Response to Reviewer 1

The manuscript by I-Ling Chen et al. entitled “Impact of Medical Conditions and Interventions on Successful Weaning from Nasal CPAP in Very Preterm Neonates: An Observational Study” focuses on an interesting topic in perinatal medicine. Particularly, authors attempted to evaluate clinical variables that could affect successful weaning from NCPAP in very preterm infants. 

Although the aims of the study are relevant, the manuscript has some major flaws.

Abstract

  • The section needs to be revised and better clarify the study design

Response: Thank you for your comment. We have revised the abstract to clarify the retrospective nature of the study design. Please see the revised abstract at line 13-27 in the revised version of the manuscript.

Method section: 

  • Material and Method section is difficult to read, especially the "study population" part. The section needs to better guide and clarify the design of the study
  • The classification of groups should be made more easily interpretable. For example could be better starting directly with the eligible patients of the study instead of infants enrolled.  

Response: We have revised the “Study Participants” section to provide inclusion and exclusion criteria. Please see this revised this in the current version of the manuscript.

Result section:

  • The result section is difficult to read and does not provide clear and concise data on the results of the study

It would be better if you would kindly revise the manuscript by a native English speaker

Response: We have extensively revised this section to ensure a native tone.

Reviewer 2 Report

The use of non invasive ventilation strategies (such as nCPAP) in the preterm infant is a current hot topic in Neonatology. Despite this, there is no evidence currently supporting one weaning method over another. This has resulted in a wide variation of practice. I therefore congratulate the authors for attempting to tackle this important, relevant question. In the current form, however, the manuscript requires some major clarification, in particular in the methods section (see below). I would also highly encourage the authors to make use of a professional editing service to ensure better flow and structure of the English language. I have made a few corrections to sentence structure/grammar (see below), but there are many other corrections still needed throughout the paper.

Title: I suggest a small change to the title, such as, : Impact of illness Severity and Interventions on Successful Weaning from Nasal CPAP in Very Preterm Neonates: an observational Study 

Abstract: No major concerns

Introduction:

Line 34-35... positive pressure to recruit collapsed alveoli. .. Since we know that CPAP does not work on totally collapsed alveoli, I suggest changing this line to... recruit semi collapsed alveoli and preventing atelectasis during expiration...

Methods:

Line 53: Change... and required positive pressure... to requiring positive pressure...

Line 62: The term birth body weight is confusing... is this the same as birth weight? If so, please correct to birth weight, otherwise please explain what you mean by birth body weight. Once again, line 64.. body weight... is this current weight? 

Line 71: Anemia... this should be anemic

Line 71: .. with the need of blood transfusions with packed red cells... suggest changing to: .. with the need for packed red cells.....

Line 72: .. <30% if under CPAP use, or <35% under intubation. Suggest changing to : <30% if infant requiring CPAP, or <35% if intubated and mechanically ventilated.

Line 79-80: If the infants have well tolerated.... suggest changing to

If the infant tolerates enteral nutrition...

Methods of nCPAP weaning.... this needs a lot of clarification. For example, do all patients start on a PEEP of 6? AT what PEEP are infants taken off CPAP? Are they weaned down to a PEEP of 5 or 4 or is CPAP discontinued from a higher PEEP. Group 1: Do all infants in this group follow the same protocol., for example 4 hrs off CPAP on day 1, 8 hrs day 2 etc. In group 2, after re-starting CPAP, how long are they on CPAP for until they are tried off again? In group 3, is there a minimum PEEP on CPAP that must be reached before HFNC can be tried?

Line 94: PMA at the successful weaning... suggest changing to PMA at the time infant was successfully weaned off CPAP.

Results: 

Line 101: Describe what you mean by an outlier. Also change (who was received a tracheostomy to... who ultimately ended up with a tracheostomy.

I am not going to make more suggestions for the results section at this time because a lot of my suggestions will depend on the clarifications in the methods section.

Tables and Figures: No major issues

Discussion:

Overall, the discussion reads well, but would benefit from an editing service to improve sentence structure/flow in the English language

Line 190: Describe what you mean by worse conditions

Line 230: Describe what you mean by medical use

Author Response

Response to Reviewer 2

The use of non-invasive ventilation strategies (such as nCPAP) in the preterm infant is a current hot topic in Neonatology. Despite this, there is no evidence currently supporting one weaning method over another. This has resulted in a wide variation of practice. I therefore congratulate the authors for attempting to tackle this important, relevant question. In the current form, however, the manuscript requires some major clarification, in particular in the methods section (see below). I would also highly encourage the authors to make use of a professional editing service to ensure better flow and structure of the English language. I have made a few corrections to sentence structure/grammar (see below), but there are many other corrections still needed throughout the paper.

Title: I suggest a small change to the title, such as, : Impact of illness Severity and Interventions on Successful Weaning from Nasal CPAP in Very Preterm Neonates: an observational Study 

Response: We have revised the title per your suggestion.

Abstract: No major concerns

Introduction:

Line 34-35... positive pressure to recruit collapsed alveoli. .. Since we know that CPAP does not work on totally collapsed alveoli, I suggest changing this line to... recruit semi collapsed alveoli and preventing atelectasis during expiration...

Response: We have revised your recommendation in red font.

Methods:

Line 53: Change... and required positive pressure... to requiring positive pressure...

Response: We have revised your recommendation in red font.

Line 62: The term birth body weight is confusing... is this the same as birth weight? If so, please correct to birth weight, otherwise please explain what you mean by birth body weight. Once again, line 64.. body weight... is this current weight? 

Response: We have revised your recommendation. Birth weight is denoted as BBW and current body weight is denoted as BW in red font.

Line 71: Anemia... this should be anemic

Response: We have revised your recommendation in red font.

Line 71: .. with the need of blood transfusions with packed red cells... suggest changing to: .. with the need for packed red cells.

Response: We have revised your recommendation in red font.

Line 72: .. <30% if under CPAP use, or <35% under intubation. Suggest changing to : <30% if infant requiring CPAP, or <35% if intubated and mechanically ventilated.

Response: We have revised your recommendation in red font.

Line 79-80: If the infants have well tolerated.... suggest changing to If the infant tolerates enteral nutrition...

Response: We have revised your recommendation in red font.

Methods of nCPAP weaning.... this needs a lot of clarification. For example, do all patients start on a PEEP of 6? AT what PEEP are infants taken off CPAP? Are they weaned down to a PEEP of 5 or 4 or is CPAP discontinued from a higher PEEP. Group 1: Do all infants in this group follow the same protocol., for example 4 hrs off CPAP on day 1, 8 hrs day 2 etc. In group 2, after re-starting CPAP, how long are they on CPAP for until they are tried off again? In group 3, is there a minimum PEEP on CPAP that must be reached before HFNC can be tried?

Response: We have provided details of the weaning method in red font. (Line 88-94)

Line 94: PMA at the successful weaning... suggest changing to PMA at the time infant was successfully weaned off CPAP.

Response: We have revised your recommendation in red font.

Results: 

Line 101: Describe what you mean by an outlier. Also change (who was received a tracheostomy to... who ultimately ended up with a tracheostomy.

Response: We have revised your recommendation in red font.

I am not going to make more suggestions for the results section at this time because a lot of my suggestions will depend on the clarifications in the methods section.

Tables and Figures: No major issues

Discussion:

Overall, the discussion reads well, but would benefit from an editing service to improve sentence structure/flow in the English language

Line 190: Describe what you mean by worse conditions

Response: We have revised to include a description of shock and sepsis as the “worse condition” in red font.

English editing was done for this manuscript.

Line 230: Describe what you mean by medical use

Response: We have made the revision to correct this term as “medication usage” at line in red font.

Reviewer 3 Report

This is an important area for neonatal research . 

Few comments - 

1- What was the rationale for eliminating the sicker infants who failed first attempt to wean and what was the denominator ?

2-There is no mention of hydrocortisone or any other type of post natal steroids that were used for extubation ? Was this not a known practice at the institution ?

3-Line 78 - "Gastric tube " is slightly misleading -suggest using  orogastric or nasogastric tube instead .

4-Consider removing heated high flow nasal canula as a weaning method if you are eliminating it from results 

5-NTISS : is it a validated scoring system , if yes please include brief description in methods 

6 - For table 2- Did you include both medical and surgical NEC or excluded the surgical NEC  . IVH - Grade 1/2  and severe IVH(3/4 ); or excluded the severe IVH 

7- Line 123 : 0.07g should be 0.07weeks 

8-Line 180-81 : Lines should be reframed regarding larger body sizes and instead use like adequate or appropriate  weight gain 

Author Response

Response to Reviewer 3

This is an important area for neonatal research. 

Few comments - 

1- What was the rationale for eliminating the sicker infants who failed first attempt to wean and what was the denominator ?

Response: Because we could not rely on clear-cut definition for the weaning periods, we exclude the sicker infants who failed at first attempt of weaning. We included 61 infants, and only one was excluded because of its failed first attempt to wean.

2-There is no mention of hydrocortisone or any other type of post natal steroids that were used for extubation ? Was this not a known practice at the institution ?

Response: We did not prefer using postnatal steroids for extubation. In this study, there were 4 infants that received dexamethasone after extubation to prevent post-extubation laryngeal edema and stridor.

3-Line 78 - "Gastric tube " is slightly misleading -suggest using orogastric or nasogastric tube instead .

Response: We have revised your recommendation in brown font.

4-Consider removing heated high flow nasal canula as a weaning method if you are eliminating it from results 

Response: We have removed this approach in the method section. Please see the revised method of weaning.

5-NTISS : is it a validated scoring system , if yes please include brief description in methods 

Response: We have provided a brief description of NTISS in the method in brown font.

6 - For table 2- Did you include both medical and surgical NEC or excluded the surgical NEC  . IVH - Grade 1/2  and severe IVH(3/4 ); or excluded the severe IVH 

Response: We include both medical and surgical NEC and any grade IVH; There were seven infants with grade 1 IVH, four infants with grade 2 IVH, one infant with grade 3 IVH, and no grade 4 IVH in this study. We have added the description for NEC and IVH in Table 2 in brown font.

7- Line 123 : 0.07g should be 0.07weeks 

Response: Thank you for your suggestion. Because the number of 0.07 is for every gram in BBW, we keep the description with 0.07 grams. (Line 130).

8-Line 180-81 : Lines should be reframed regarding larger body sizes and instead use like adequate or appropriate weight gain 

Response: We have revised your recommendation to “adequate weight gain” in brown font.

Round 2

Reviewer 2 Report

Regarding specific edits, the authors made all relevant changes and these sections read better now. However, the paper would still benefit from an editing service to improve sentence structure / flow of the English language. I made some of the corrections, but there are still plenty more!
